

# Detection-confirmation-standardization-quantification: a novel method for the quantitative analysis of active components in traditional Chinese medicines

Xiaohong Song*, Anyi Zhao*, Yan Liu, Jintang Cheng, Sha Chen and An Liu

Key Laboratory of Beijing for Identification and Safety Evaluation of Chinese Medicine, Institute of Chinese Materia Medica, China Academy of Chinese Medical Sciences, Beijing, China
* These authors contributed equally to this work.

## ABSTRACT

**Background:** Quantitative analysis of the active ingredients of Traditional Chinese Medicine is a research tendency. The objective of this study was to build a novel method, namely, Detection-confirmation-standardization-quantification (DCSQ), for the quantitative analysis of active components in traditional Chinese medicines, without individual reference standard.

**Methods:** Danshen (the dried root of Radix *Salvia miltiorrhiza*) was used as the matrix. The "extraction" function in high-performance liquid chromatography-mass (HPLC-MS) instrument was used to find the peaks corresponding to cryptotanshinone, tanshinone I, and tanshinone IIA in the total ion current (TIC) chromatogram of Danshen. The multicomponent reference standard (MCRS) containing the three tanshinones mainly was prepared by preparative HPLC. The contents of them in the resulting MCRS were determined by NMR; moreover, the constituents of the MCRS were confirmed. The MCRS containing known content of the three tanshinones was used as the reference standard for the quantitative analysis of cryptotanshinone, tanshinone I and tanshinone IIA in Danshen Samples by analytical HPLC.

**Results:** The optimized HPLC conditions for the quantitative analysis of the active components in Danshen were established, and the assignments of the extracted peaks were confirmed by analyzing the characteristic fragments in their MS/MS product ion spectra and UV spectra. Then, the MCRS containing the three tanshinones were successfully prepared. The results of determination about the contents by NMR showed linearity fitted with high likelihood and calibration curves possessed high linearity. The results of determination on Danshen Samples obtained through DCSQ exhibited minimal deviations, in contrast to those obtained through individual reference standards.

**Conclusion:** The establishing DCSQ is independent and convenient for the quantitative analysis of the active components in TCMs by MCRS, without individual reference standard. This method is a great advance in quantitative analysis for complex composition, especially TCMs.

Corresponding authors
Sha Chen, schen@icmm.ac.cn
An Liu, aliu@icmm.ac.cn

# INTRODUCTION

Traditional Chinese medicines (TCMs) have been widely used in the treatment of various diseases because of their remarkable and reliable biological activities. Therefore, the determination of the active components in TCMs by chromatography methods such as high-performance liquid chromatography (HPLC), thin-layer chromatography (TLC), and high-performance capillary electrophoresis (HPCE) are considered as the main strategy by which the quality of TCMs may be controlled. Conventionally, the active components of TCMs are quantified by the corresponding reference standards (*National Commission of Chinese Pharmacopoeia, 2020a*). Popular compounds are purchased from authoritative organizations, whereas rare compounds are purified inhouse. Another approach, namely, single standard to determine multicomponents (SSDMC), had been developed to reduce the reliance of quantity on the reference standards (*Fang et al., 2017*; *Liu et al., 2017*). The reference standards are still needed when assigning peaks and calculating conversion factors (the molar response ratio of reference standard to analyte).

NMR spectroscopy is considered as a promising quantification method for the direct determination of target compounds in mixtures without reference standards; the amount of analytes can be calculated using the ratios of the signal intensities of the protons of different compounds and internal reference standard (*Frezza et al., 2018*; *Luisa, Maria & Garcia, 2017*; *Petrakis et al., 2017*; *Chauthe et al., 2012*; *Staneva et al., 2011*). However, a major challenge associated with NMR is the difficulty involved in the quantification of complex mixtures. The NMR spectra of complex mixtures often exhibit severe peak overlap, thereby affecting the accuracy of the quantity of analyte. To prevent severe peak overlap, *Staneva et al. (2011)* fractionated the extracts of TCMs before quantifying the target components by NMR (*Chauthe et al., 2012*; *Staneva et al., 2011*). This procedure often leads to the dissociation of target components into separate parts or absorption by sorbents. Consequently, the results do not accurately reflect the contents of target compounds in TCMs.

We proposed a novel method for the quantitative analysis of the active components in TCMs without individual reference standards, namely, detection-confirmation-standardization-quantification (DCSQ). Danshen (the dried root of *Salvia miltiorrhiza*), which was a popular traditional Chinese medicinal herb best known for its putative cardioprotective and anti-atherosclerotic effects, was used as the matrix (*Jia et al., 2019*; *Liu et al., 2015*; *Wang et al., 2011*). The main components responsible for its pharmacological properties were hydrophilic depsides and lipophilic diterpenoid quinines (*Wang et al., 2013*; *Li, Xu & Liu, 2018*). An application of the DCSQ method for the quantitative analysis of three main diterpenoid quinones, cryptotanshinone, tanshinone I, and tanshinone IIA (Figs. 1A–1C) was reported for the first time.

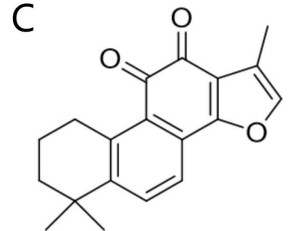

**Figure 1 Chemical structures of cryptotanshinone (A), tanshinone I (B), tanshinone IIA (C), and dimethyl fumarate (D).**

HPLC is a suitable technique for the quantitative analysis of the active components in TCMs because they contain numerous compounds. The separation of target components from a complex mixture in TCMs can be achieved by optimizing the HPLC conditions. Based on the combination of HPLC, MS, and NMR techniques, the quantitative analysis of the target components in TCMs without reference standard can be performed as follows: First, the "extraction" function in HPLC-MS instrument was used to find the peaks corresponding to the target components from the complex TIC chromatogram of TCMs, even without adequate separation (*Wang et al., 2017*). Next, the peaks were confirmed by the analysis of their MS/MS spectra. Finally, the HPLC conditions were optimized for the quantitative analysis of target components. The mixture mainly consisting of the target components, referred to as the multicomponent reference standard (MCRS), was prepared by preparative HPLC. The contents of the target components in the MCRS were determined directly by NMR, and the MCRS was used as the reference standard instead of individual reference standards.

## MATERIALS & METHODS

### Materials and chemicals

Danshen was purchased from the Qiancao herbal wholesale company (Beijing). The standards of cryptotanshinone (1), tanshinone I (2), and Tanshinone IIA (3) (Figs. 1A–1C) were obtained from Traditional Chinese Medicine Solid Preparation National Engineering Research Center (Nangchang, Jiangxi Province, China), with a purity of ≥98%. Dimethyl fumarate (4) (Fig. 1D) with a purity of 99% was obtained from Alfa Aesar (Ward Hill, MA, USA). CDCl₃ (99.8% pure) was obtained from Cambridge Isotope Laboratories Inc. (Andover, MA, USA). Acetonitrile and methanol for HPLC were obtained from Fisher Scientific (Fair Lawn, NJ, USA). Formic acid (>98% purity) for HPLC was obtained from Sinopharm Chemical Reagent Co. Ltd. (Shanghai, China). All other chemicals were of analytical grade. Deionized water was obtained from Wahaha Company (Hangzhou, Zhejiang, China).

## Identification of cryptotanshinone, tanshinone I, and tanshinone IIA in Danshen by HPLC-ESI-MS/MS

Sample preparation: Dried Danshen samples were powdered using a mill and sieved through a No. 24 mesh. Approximately 0.3 g of the powder sample was weighed and extracted by refluxing in 50 mL of methanol for 1 h. The weight loss was adjusted with methanol after the extraction. One mL of the sample solution was filtered through a 0.45 μm nylon filter into a HPLC amber sample vial for injection.

Data acquisition: An Agilent 1,100 series HPLC system (Agilent Technologies, Santa Clara, CA, USA) equipped with a quaternary solvent delivery system, an on-line degasser, an auto-sample injector, a column temperature controller, and a variable wavelength detector (VWD) was coupled to an Agilent 6,460 triple quadruple mass spectrometer (QQQ-MS) equipped with a dual electrospray ion source (ESI) (Agilent Technologies, Santa Clara, CA, USA) formed the HPLC-ESI-MS/MS system.

The samples were separated using a Diamonsil C18 column (4.6 × 250 mm, 5 μm, Dikma Technologies Inc., Foothill Ranch, CA, USA). The mobile phase was a mixture of acetonitrile (mobile phase A) and water containing 0.1% formic acid (mobile phase B). The gradient elution started at 60% A, followed by a linear increase to 80% A at 30 min, 90% A at 40 min, a linear decrease to 60% A at 41 min, and held constant at 55 min before the next injection was performed. The flow rate was 1.0 mL min$^{-1}$. The column temperature was maintained at 22 °C. The injection volume was 20 μL.

The mass spectrometric data were acquired using the positive electrospray mode. The full-scan mass spectrum was recorded over the range of *m/z* 250-325. $N_2$ was used as the sheath and auxiliary gas of the mass spectrometer. The capillary voltage for ESI spectra was 3.5 kV, and the capillary temperature was set at 300 °C. Ultra-high pure helium was used as the collision gas in the collision-induced dissociation (CID) experiments. The MS/MS product ion spectra were acquired through the CID of the peaks with the protonated molecular ion [M + H] $^+$ of each analyte. The collision energy for the target protonated molecular ions was set at 20 eV to obtain the appropriate fragment information.

## Preparation of multicomponent reference standard

Preparation of total tanshinones: The total tanshinones were prepared using a method of preparing tanshinone extract as recorded in the *National Commission of Chinese Pharmacopoeia (2020b)* edition. Fifty gram of the Danshen powder was extracted by refluxing in 500 mL 95% ethanol for 2 h. The Danshen extract was evaporated under reduced pressure at 40 °C. The resulting solid was washed three times with hot water (80 °C) to remove the water-soluble components and obtain the total tanshinones.

Preparation of multicomponent reference standard (MCRS): the MCRS containing the three tanshinones was prepared using a CXTH LC-3000 semi-preparative HPLC series equipped with a binary solvent delivery system, a Rheodyne 7725i manual injection valve (five mL sample loop) and a UV-visible detector (Chuangxintongheng Co. Ltd., Beijing, China). The total tanshinones were dissolved in 30 mL of methanol and separated using a Thermo BDS Hypersil C18 prep-column (22.2 × 150 mm$^2$, 5 μm, Thermo Scientific,

Waltham, MA, USA) eluting with methanol/water (77:23, v/v). The flow rate was 9.0 mL min$^{-1}$, and the detection wavelength was set at 200 nm. The injection volume was 1.0 mL. All the effluents containing cryptotanshinone, tanshinone I, and tanshinone IIA were collected and evaporated to dryness. A total of 210 mg of the MCRS was obtained from 30 mL of the total tanshinone solution.

## Quantitative determination of each tanshinones in multicomponent reference standard by QNMR

Generation of NMR calibration series: a series of volumetric solutions containing 20.50, 10.25, 6.83, 3.42 and 1.71 mg/mL of the MCRS and 2.96, 1.48, 0.74, 0.37 and 0.19 mg/mL of dimethyl fumarate in CDCl$_3$ were prepared and measured by NMR at 600 MHz.

NMR spectral acquisition and processing parameters: The spectra was acquired using a 14.1 T Bruker Avance 600 MHz NMR spectrometer equipped with a 5 mm broad band (BB) inverse detection probe tuned to detect $^1$H resonances. The $^1$H resonance frequency was 600.13 MHz. All the spectra were acquired at 298 K. A total of 64 scans of 32 K data points were acquired with a spectral width of 9615.4 Hz (16 ppm). A pre-acquisition delay of 6.5 μs, with an acquisition time of 1.7 s, recycle delay of 24 s, and flip angle of 90° were used. The chemical shift of all the peaks was referenced to the tetramethylsilane (TMS) resonance at 0 ppm. The spectra were Fourier transformed to afford a digital resolution in the frequency domain of 0.293 Hz/Point. The phase and baseline corrections of the spectra were carried out manually. Preliminary data processing was carried out using the Bruker software TOPSPIN 2.1.

The signals for the H-15 of cryptotanshinone (4.884 ppm, t, 2H), H-17 of tanshinone I (2.295 ppm, d, 3H), H-17 of tanshinone IIA (2.259 ppm, d, 3H), and olefinic H of dimethyl fumarate (6.864 ppm, s, 2H) were used to determine the contents of cryptotanshinone, tanshinone I, and tanshinone IIA in the MCRS.

The concentrations of the three tanshinones in the MCRS were calculated using the following general equation:

$$C_x = \frac{A_x \times W_i \times N_i}{A_i \times M_i \times N_x \times V} \tag{1}$$

where $C_x$ (in mM) corresponds to the concentrations of the three individual tanshinones; $A_x$ and $A_i$ correspond to the peak areas of tanshinones and the internal standard. $W_i$ corresponds to the mass of the internal standard (in mg); $N_i$ and $N_x$ correspond to the number of protons of the respective signals of the internal standard and tanshinones used for the quantitative analysis; $M_i$ corresponds to the molecular weight (in Da) of the internal standard, and $V$ (in L) corresponds to the volume of CDCl$_3$.

The contents of the three tanshinones in the MCRS were calculated using the following general equation:

$$P_x = \frac{C_x \times M_x \times V}{W_m} \times 100\% \tag{2}$$

where $P_x$ (%) corresponds to the percentage of the three individual tanshinones in the MCRS; $M_x$ corresponds to the molecular weight (in Da) of the three tanshinones, and $W_m$ corresponds to the mass of the MCRS (in mg).

## Quantitative analysis of the three tanshinones in the Danshen sample by HPLC using the MCRS as the reference standard

Sample preparation and HPLC conditions: The Danshen sample was prepared according to the procedure described according to the above steps.

Analytical HPLC was carried out using a Shimadzu LC-20AT series equipped with a quaternary solvent delivery system, an on-line degasser, and auto-sample injector, a column temperature controller, and an SPD-M20A diode-array detector (DAD) (Shimadzu corporation, Kyoto, Japan). Finally, the samples were separated using a Diamonsil C18 column (4.6 × 250 mm, 5 μm, Dikma Technologies Inc.). Notably, the HPLC conditions are compatible to the HPLC-ESI-MS experiments.

Construction of calibration curves: Appropriate amounts of the MCRS were weighed and dissolved into 100 mL of acetone. To construct the calibration curves, 2, 4, 8, 12, 16 and 20 μL of the solution were injected in triplicates. The calibration curves were constructed by plotting the peak areas versus the quantity of the three tanshinones in the MCRS.

## RESULTS

### Detection of the peaks corresponding to cryptotanshinone, tanshinone I, and tanshinone IIA from the TIC and HPLC chromatograms of Danshen

The initial investigation focused on determining the target components from the TIC and HPLC chromatograms of Danshen and establishing the optimized HPLC conditions suitable for the quantitative analysis of the active components in Danshen. A rough gradient elution was first performed, after which the peaks corresponding to cryptotanshinone, tanshinone I, and tanshinone IIA were searched with their protonated molecular ion $[M + H]^+$ from the total ion chromatogram of Danshen (*Wang et al., 2017*). The results (Figs. 2A and 2C) indicated that the peak with the retention times of 6.4, 6.2 and 8.5 min corresponded to the *m/z* of 297, 277 and 295, respectively. Under these conditions, cryptotanshinone and tanshinone I were not adequately separated, even though the positions of cryptotanshinone and tanshinone I in the TIC and HPLC spectra of Danshen were successfully assigned. The result obtained by the optimization of gradient elution is shown in Figs. 2B and 2D; the peaks with the retention times of 17.2, 19.0 and 26.1 min corresponded to the *m/z* of 297, 277, and 295, respectively, in the TIC of LC-MS (Fig. 2B). Moreover, the three tanshinones were adequately separated from other compounds. This condition was adequate for the subsequent quantitative analysis of the three tanshinones in Danshen.

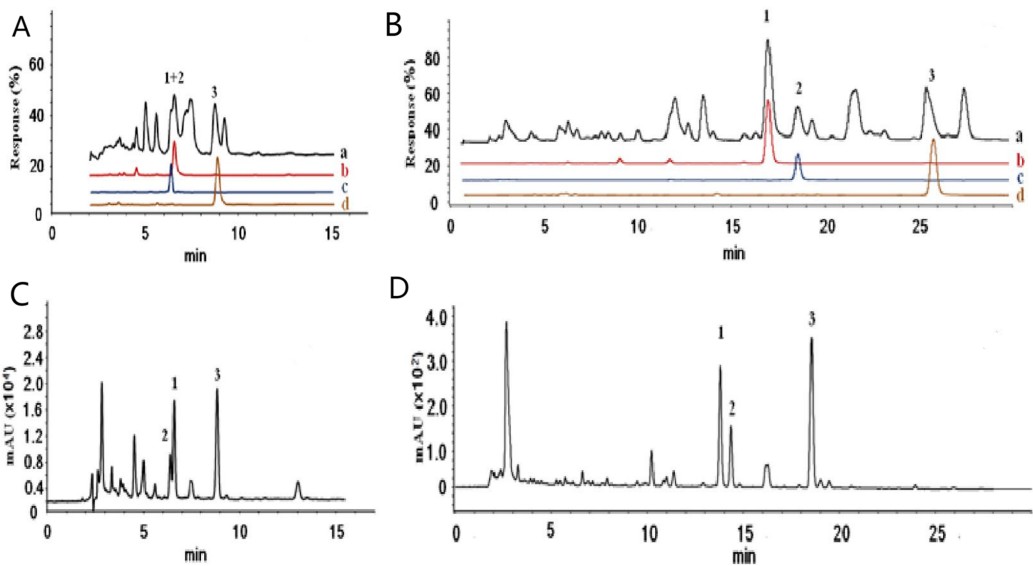

**Figure 2 Total ion current chromatograms of the Danshen extract and extracted ion chromatograms of target protonated molecular ions (A–B) and UV spectrum of the Danshen extract (C–D).** The HPLC conditions were as follows: The water phase contained 0.1% formic water. The flow rate was 1.0 mL/min. The detection wavelength was set at 270 nm. The eluting condition of (A–B) was as follows: 85% acetonitrile was maintained for 15 min, and the column temperature was 35 °C. The eluting condition of (C–D) was as follows: 60% acetonitrile was maintained for 0–30 min, and the concentration of acetonitrile was increased from 60% to 80% and then to 90% of acetonitrile at 40 min. The column temperature was 22 °C.           

## Confirmation of the peak assignments by ESI-MS/MS product ion and UV spectra

The ESI-MS/MS product ion spectra were acquired through the CID of the peaks obtained with the protonated molecular ion $[M + H]^+$ of the three tanshinones. The proposed MS fragmentation pathway of the compounds with the protonated molecular ions of *m/z* 297, 277 and 295 were summarized in Fig. 3, consistent with the MS fragmentation pathway of cryptotanshinone, tanshinone I, and tanshinone IIA as studied by *Wang et al. (2017)*. The UV spectra shown in Fig. 4 were obtained directly from the HPLC experiments. The UV spectra of the three peaks matched with those of cryptotanshinone, tanshinone I, and tanshinone IIA, as reported in the current literature (*Huang et al., 2018*).

## Preparation of the MCRS containing mainly cryptotanshinone, tanshinone I, and tanshinone IIA

The MCRS containing mainly cryptotanshinone, tanshinone I, and tanshinone IIA was prepared by the preparative HPLC of the total tanshinone to perform a quantitative analysis of the three components, without individual reference standards. The chromatograms of the total tanshinone and MCRS were shown in Fig. 5A. Peaks 1, 2, and 3 were assigned to cryptotanshinone, tanshinone I, and tanshinone IIA, respectively, by collecting the main peaks in the preparative HPLC separately and then analyzing them by analytical HPLC.

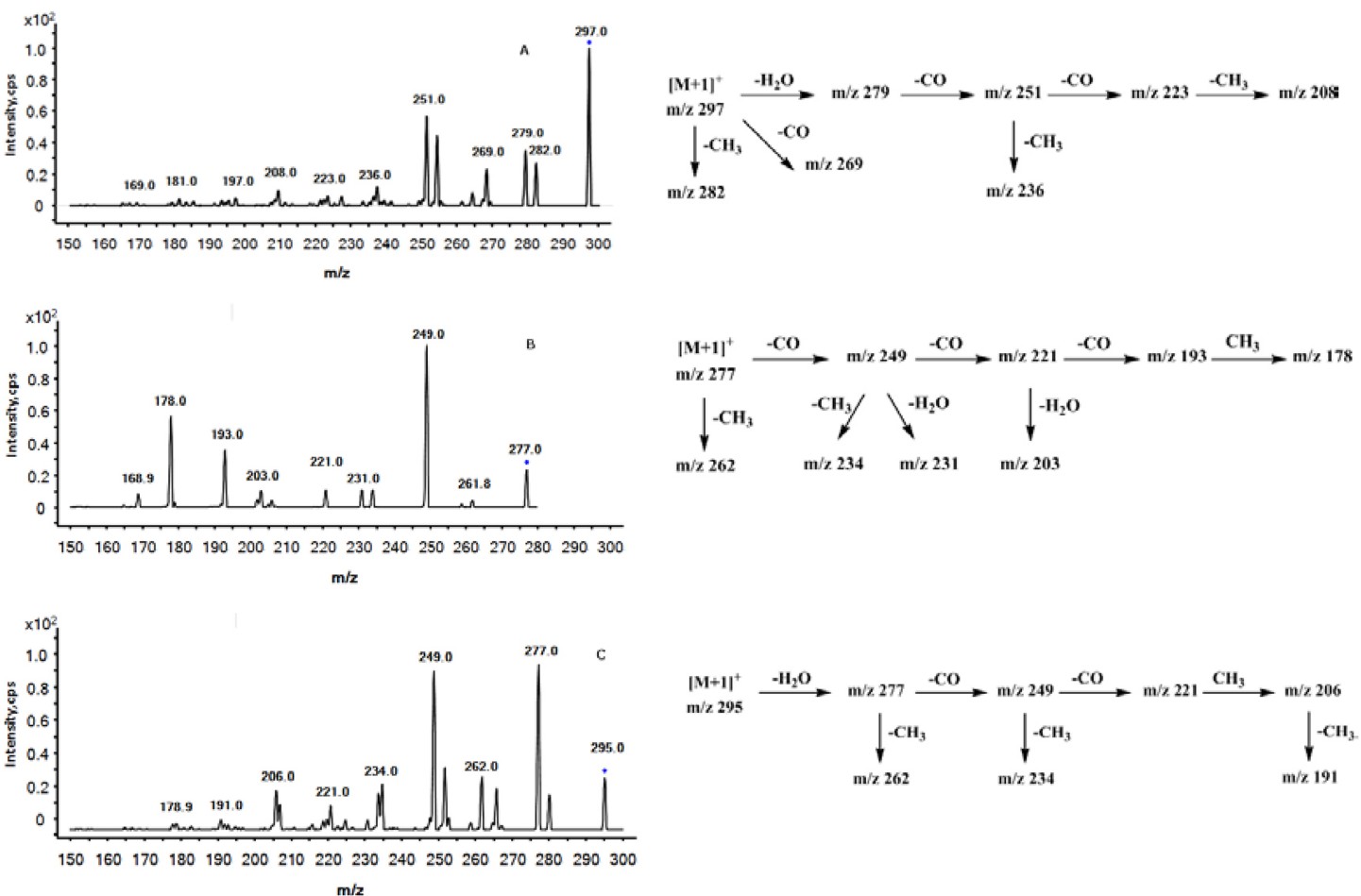

**Figure 3 Positive ion ESI-MS/MS spectra and the proposed MS fragmentation pathways of peaks 1 (A), 2 (B) and 3 (C).**

## Determination of the contents of cryptotanshinone, tanshinone I, and tanshinone IIA in the MCRS by NMR

The contents of the three tanshinones in the resulting MCRS were determined directly by $^1$H NMR analyses. The $^1$H NMR spectrum of the MCRS mixed with the internal standard, dimethyl fumarate, was shown in Fig. 6. The main signals were assigned to the three tanshinones according to the references, excluding the signals belonging to dimethyl fumarate (*Mei et al., 2019*; *Zeng et al., 2017*; *Wu, Jiang & Wu, 2015*). The composition of the MCRS was confirmed again by analyzing the signals observed in the $^1$H NMR spectrum. To perform an accurate quantitative analysis, the data processing technique was used which named the Lorentz deconvolution function in the Bruker software TOPSPIN 2.1 . The linearity of signals were fitted by the Lorentz method. Next, the deconvolution was performed, and the peak areas were automatically generated. Regarding the quantitative analysis of the mixtures by NMR using the data processing, the accuracy was determined by the likelihood of lineshape fitting. The results showed that

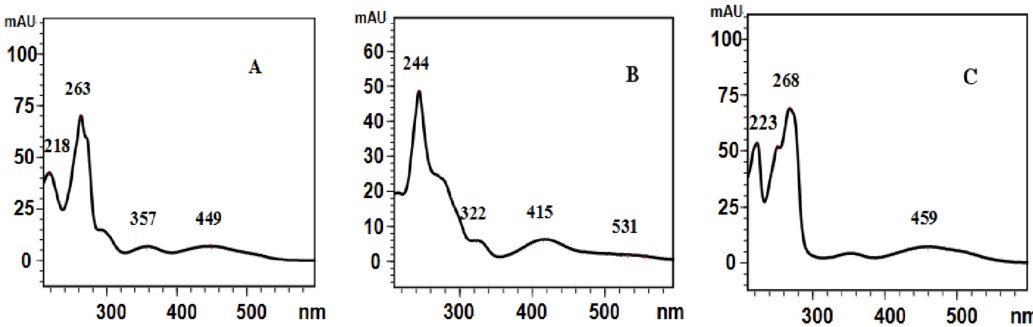

**Figure 4 UV spectra of the three peaks.** UV spectra of the three peaks shown in Fig. 2D. (A) UV spectrum of the peak with the retention time of 16.8 min; (B) UV spectrum of the peak with the retention time of 18.6 min; (C) UV spectrum of the peak with the retention time of 25.7 min.

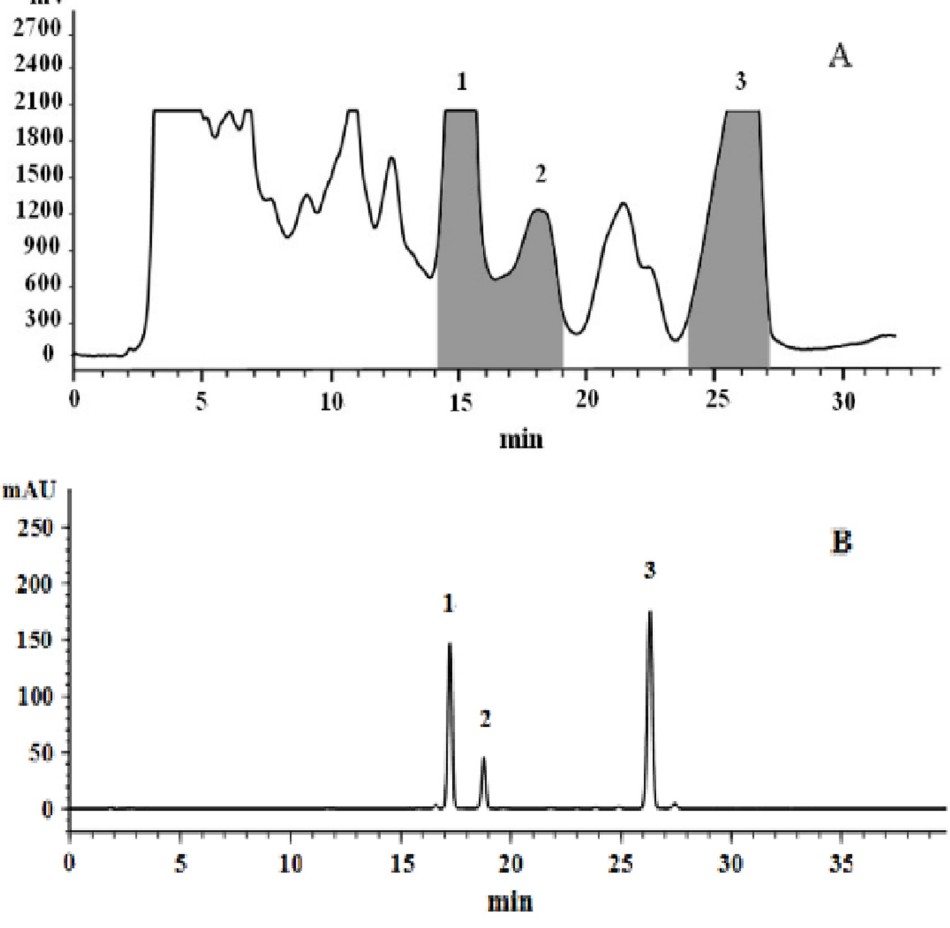

**Figure 5 Representative preparative HPLC chromatogram of total tanshinones (A) and analytical HPLC chromatogram of the MCRS (B).** The MCRS was analyzed under the same condition as HPLC–MS experiment (eluting gradient of Figs. 2C–2D).

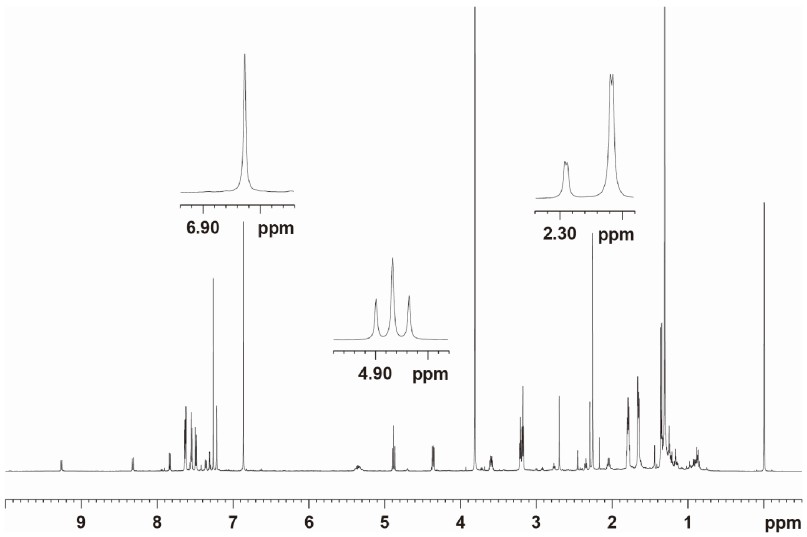

**Figure 6 600 MHz $^1$H NMR spectrum of the MCRS mixed with dimethyl fumarate.** The concentrations of the MCRS and dimethyl fumarate in CDCl3 were 20.50 and 2.96 mg·mL$^{-1}$, respectively.

the contents of crytotanshinone, tanshinone I, and tanshinone IIA were at 1.635 g/100 g, 0.718 g/100 g, 1.953 g/100 g in Danshen sample, respectively, by MCRS method.

Under the preparation of MCRS of Danshen, the selection of the signals in the NMR spectra for the quantitative analysis of the three tanshinones and the validity of the NMR method were performed. The analytical method of NMR was achieved a high separation efficiency which was shown in Figs. 7A–7O. The NMR spectrum is very reproducible with good reproducibility and target three compounds high separation efficiency.

## Determination of the contents of the three tanshinones in the Danshen sample by HPLC using the MCRS as the reference standard and comparison of the results with those obtained using individual reference standards

The calibration curve of the MCRS was constructed to investigate the validity of the NMR method. The calibration curves were constructed by plotting the concentrations of the three tanshinones versus that of dimethyl fumarate. Next, the linear regression lines were calculated. The calibration graph demonstrating the linearity of the NMR response with increasing concentrations of the three tanshinones was shown in Fig. 8.

Approximately 5 mg of the MCRS was dissolved into 100 mL of acetone to construct the calibration curve of the three tanshinones, corresponding to ~1.5, 0.5 and 1.7 mg of individual cryptotanshinone, tanshinone I, and tanshinone IIA, respectively. The contents of the three tanshinones in the Danshen sample were determined according to the calibration curves based on individual standards. The content of cryptotanshinone was determined at 1.635 g/100 g and 1.627 g/100 g, respectively, using MCRS and standard quantitation. Tanshinone I was detected at 0.718 g/100 g and 0.727 g/100 g,

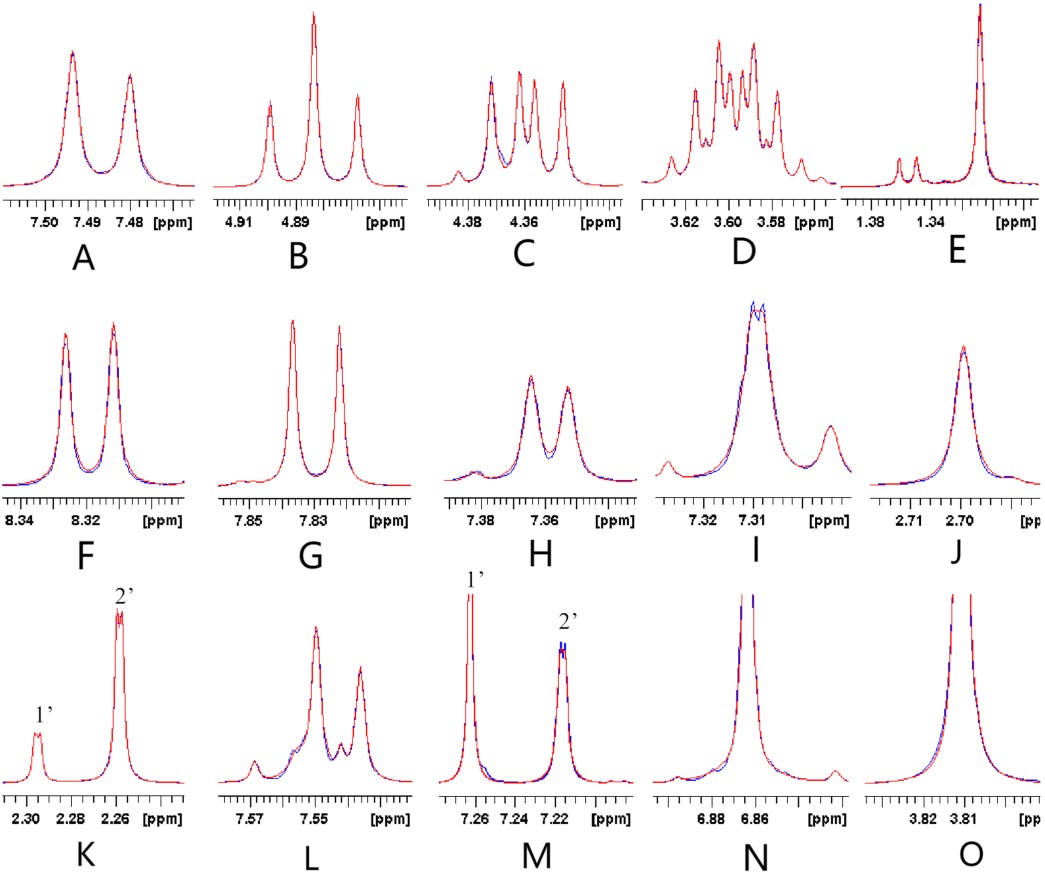

**Figure 7 Representative results of the lineshape fitting of the signals belonging to the three tanshinones and dimethyl fumarate performed by the Lorentz deconvolution.** A–E reflect the representative lineshape fitting of signals belonging to cryptotanshinone. F–K (1') reflect the representative lineshape fitting of signals belonging to tanshinone I. K (1'), L and M (2') reflect the representative lineshape fitting of signals belonging to tanshinone IIA. N and O reflect the lineshape fitting of signals belonging to dimethyl fumarate.

respectively, based on the two methods. Tanshinone IIA was measured at 1.953 g/100 g and 1.886 g/100 g, respectively. The mean contents of the three compounds obtained by the two methods were tested for no significance difference at the $P \leq 0.05$ level. The results were compared with those obtained using individual reference standards. The deviations of the three tanshinones determined by the two methods were minimal (Fig. 9), and this could be attributed to the maximum likelihood lineshape fitting of their signals in the NMR spectrum of the MCRS.

## DISCUSSION

The multicomponent reference standard (MCRS) of Danshen was preparing successfully to use for the samples' quantitative analysis. The MCRS containing cryptotanshinone, tanshinone I, and tanshinone IIA was obtained by collecting peaks 1, 2 and 3 together (Fig. 5B). The preparation of MCRS was found to be much easier than that of individual reference standards. The presence of some other compounds whose NMR signals did not overlap severely with those of the target components may be allowed. And the target

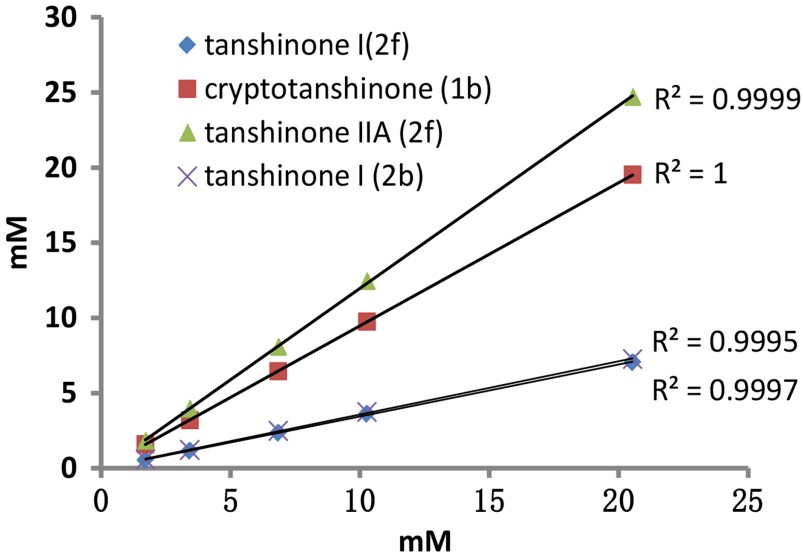

**Figure 8 Concentrations of the three tanshinones determined by specific signals.**

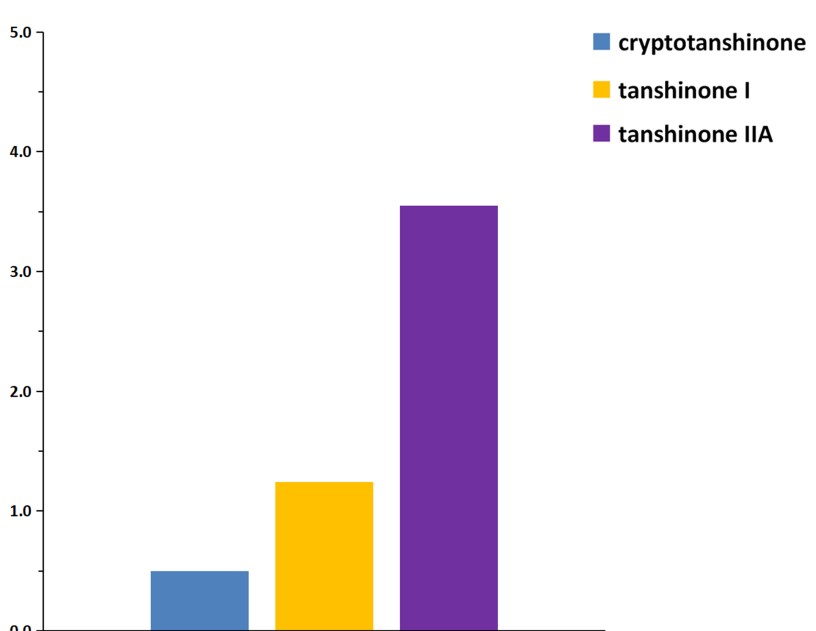

**Figure 9 Deviations of the contents of the three tanshinones in the Danshen sample as determined by the MCRS contrast to individual reference standards.**

components need not be separated from each other in their preparative HPLC spectrum because they were finally collected for preparing the MCRS.

Apart from structure elucidation, NMR had been progressed to become a useful technique for the direct quantitative analysis of complex mixtures. However, it remained difficult to directly quantitatively analyze the target components from the extracts of

TCMs using the NMR technique, owing to the existence of numerous components, which causing severe overlap of the NMR signals. In this study, HPLC was selected for the quantification of the three tanshinones in Danshen for its outstanding separation ability compared to NMR. By using the "extraction" function in HPLC-MS instrument to find the target components and MCRS as the reference standard diminished the dependency of individual reference standards in the quantitative analysis of the target components in TCMs. Evidently, the increased amount of components in TCM samples would not make the quantitative analysis of target components more difficult, which could be attributed to the efficient separation function of HPLC. The components whose signals overlap with each other could be divided into different MCRSs. The amount of MCRS prepared from one sample could be adjusted according to the components requiring quantification and considering the prevalence of NMR signal overlapping.

The results of MCRS shown in Fig. 7 indicated that the extents of the likelihood of the lineshape fitting of the signals for the three tanshinones performed by the Lorentz deconvolution are different. Figs. 7A–7E, 7F–7K, 7L–7M and 7N–7O showed the lineshape fitting results of the signals for cryptotanshinone, tanshinone I, tanshinone IIA, and dimethyl fumarate, respectively. The lineshapes of the following signals were fitted with relatively higher likelihood: the triplet in Fig. 7B at 4.884 ppm belonged to cryptotanshinone, the doublets in Fig. 7G and Fig. 7K (1') belonged to tanshinone I, the doublet in **K (left side)** belongs to tanshinone IIA, and the singlet in **N** belonged to dimethyl fumarate.

The calibration graph demonstrating the linearity of the NMR response with increasing concentrations of the three tanshinones in Fig. 8 indicated that the $R^2$ values calculated from the signals in Fig. 7C and Fig. 7K belonging to cryptotanshinone and tanshinone IIA were 1.0000 and 0.9999, respectively. The signals in Fig. 7G and Fig. 7K (2') belonging to tanshinone I also exhibited a high likelihood of lineshape fitting, and the $R^2$ values calculated from the two signals were 0.9995 and 0.9997, respectively. The result confirmed that the concentrations of the three tanshinones in the MCRS could be accurately determined using $^1$H NMR after deconvolution. The average contents of three tanshinones was calculated with five different concentrations from the NMR spectra of the MCRS. And the results were 28.01%, 9.43% and 34.36%, respectively. The signals in Fig. 7B, Fig. 7K and Fig. 7N were finally selected for the quantitative analysis of the three tanshinones, because their linearity fitted with high likelihood and their calibration curves possessed high linearity.

The calibration curves of the three tanshinones were constructed using the same method as that of individual reference standards. A significant advantage of using the MCRS as the reference standard was that it reduced the dependency of quantitative analysis on the individual reference standard and also reduced the consumption of reference standards.

## CONCLUSIONS

In this study, the advantageous combination of HPLC, MS, and NMR techniques facilitated the accurate quantification of the target components in TCMs, without

individual reference standards, which could be defined as Detection-confirmation-standardization-quantification (DCSQ). By combination of multiple analytical techniques, the MCRS was detected, confirmed, prepared and determinated successfully. In the study, NMR' direct quantitative analysis was utilized to the MCRS even though the resolution of HPLC was not enough in the mixture. The MCRS of danshen with three major constituents, cryptotanshinone, tanshinone I, and tanshinone IIA, was prepared and used for samples. We provided an innovative method to get reference standards.

The MCRS could be a novel approach as standards for quantification about corresponding samples, which was less dependent than individual reference standards. The MCRS as the reference standard will be used to quantitative analysis and for the more industrial applications. It will be more accurate and convenient for the target analyte. It was a great advance in quantitative analysis for complex composition, especially TCMs.

### Funding

This study is supported by the Major Scientific and Technological Special Project for "the Fundamental Research Funds for the Central public welfare research institutes" (No. ZZ13-YQ-057) and "Significant New Drugs Creation" (2018ZX09735-002) and (2019ZX09201005-002-001). The funders had no role in study design, data collection and analysis, decision to publish, or preparation of the manuscript.

### Grant Disclosures

The following grant information was disclosed by the authors:
Major Scientific and Technological Special Project for "the Fundamental Research Funds for the Central public welfare research institutes": ZZ13-YQ-057.
"Significant New Drugs Creation": 2018ZX09735-002 and 2019ZX09201005-002-001.

### Competing Interests

The authors declare that they have no competing interests.

### Author Contributions

- Xiaohong Song performed the experiments, analyzed the data, performed the computation work, authored or reviewed drafts of the paper, and approved the final draft.
- Anyi Zhao performed the experiments, analyzed the data, performed the computation work, authored or reviewed drafts of the paper, and approved the final draft.
- Yan Liu performed the experiments, analyzed the data, performed the computation work, prepared figures and/or tables, and approved the final draft.
- Jintang Cheng performed the experiments, analyzed the data, performed the computation work, prepared figures and/or tables, and approved the final draft.
- Sha Chen conceived and designed the experiments, analyzed the data, authored or reviewed drafts of the paper, and approved the final draft.

- An Liu conceived and designed the experiments, analyzed the data, authored or reviewed drafts of the paper, and approved the final draft.

## Data Availability

Raw measurements are available in the Supplemental Files.

## Supplemental Information

Supplemental information for this article can be found online at http://dx.doi.org/10.7717/peerj-achem.10#supplemental-information.

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
