# Peer review of "Detection-confirmation-standardization-quantification: a novel method for the quantitative analysis of active components in traditional Chinese medicines"

_PeerJ Analytical Chemistry, doi:10.7717/peerj-achem.10_

## Round 0.1 · original submission · Major Revisions

· Academic Editor

Major Revisions

According to the Reviewers' comments, the manuscript requires a major revision before it can be considered for publication. It is important to revise the language. The English needs to be improved due to the ambiguous expression, improper expression, and grammar mistakes. Please revise the whole manuscript carefully and try to avoid any grammar or syntax errors before submission.

Reviewer 1 ·

Basic reporting

The English language should be improved to ensure that an international audience can clearly understand your text. Some examples where the language could be improved include lines 292, 322 – the word ‘reliability’ makes comprehension difficult, suggest replacing with’dependency’.

Lines 91- Is it necessary the word ‘directly’ because the preparative HPLC is different from the above-mentioned quantitative HPLC.

Please check:
Lines 169- The signals for the H-15 of cryptotanshinone (4.884 ppm, t, 1H), 1H or 2H?

Lines 218- times of 17.2, 19.0, and 26.1 min, It doesn't match the coordinates in Fig. 2 (A2 and B2);

Lines 304- the doublets in 2b and 2f (left side) belonged to 305 tanshinone I, the doublet in 2f (right side) belongs to tanshinone IIA, there is a contradiction with the annotation of Figure 7

Lines 309- in 1c and 2f (right side) (Fig. 7) 1c or 1b?

Lines 315- tanshinone IIA calculated from the NMR spectra of the MCRS with five different concentrations, five or three?

Experimental design

no comment

Validity of the findings

no comment

·

Basic reporting

The paper reported a novel strategy for the quantitative analysis of active components in traditional Chinese medicines, without individual reference standard. The proposed strategy was used for the quantitative analysis of cryptotanshinone, tanshinone I, and tanshinone IIA in Danshen Samples. I think it is worthy research work and has potentials in traditional Chinese medicines. However, there are still some details and issues as follows that need further clarification and discussion. Thus, I recommend the acceptance of this work after major revisions.

1.Accurate quantitative analysis of target analytes in complex matrices is very important, so, quantitative results of cryptotanshinone, tanshinone I, and tanshinone IIA in Danshen samples should be provided in the text.
2.The NMR spectra of complex mixtures often exhibit severe peak overlap, which can affect the accuracy of the quantity of analyte. The separation degree of the peaks in 1b, 2f (Figure 7) and 4a should be calculated.
3.The experiment process seems a little complicated, whether this will cause the loss of the target analyte?
4.The results of the proposed method should be verified by using individual reference standards and evaluated by using statistical methods.

Experimental design

sss

Validity of the findings

sss

Additional comments

The paper reported a novel strategy for the quantitative analysis of active components in traditional Chinese medicines, without individual reference standard. The proposed strategy was used for the quantitative analysis of cryptotanshinone, tanshinone I, and tanshinone IIA in Danshen Samples. I think it is worthy research work and has potentials in traditional Chinese medicines. However, there are still some details and issues as follows that need further clarification and discussion. Thus, I recommend the acceptance of this work after major revisions.

1.Accurate quantitative analysis of target analytes in complex matrices is very important, so, quantitative results of cryptotanshinone, tanshinone I, and tanshinone IIA in Danshen samples should be provided in the text.
2.The NMR spectra of complex mixtures often exhibit severe peak overlap, which can affect the accuracy of the quantity of analyte. The separation degree of the peaks in 1b, 2f (Figure 7) and 4a should be calculated.
3.The experiment process seems a little complicated, whether this will cause the loss of the target analyte?
4.The results of the proposed method should be verified by using individual reference standards and evaluated by using statistical methods.

---

## Round 0.2 · Minor Revisions

· Academic Editor

Minor Revisions

It is important to revise the language. The English needs to be improved due to the ambiguous expression, improper expression and grammar mistakes. Please revise the whole manuscript carefully and try to avoid any grammar or syntax errors before submission.

---

## Round 0.3 · accepted · Accept

· Academic Editor

Accept

Thank you for taking the time to revise your work.